# Quantum Probability’s Algebraic Origin

**DOI:** 10.3390/e22111196

**Published:** 2020-10-23

**Authors:** Gerd Niestegge

**Affiliations:** Freelance Researcher, 48683 Ahaus, Germany; gerd.niestegge@web.de

**Keywords:** quantum mechanics, probability, quantum logic, uncertainty relation, Bell-Kochen- Specker theorem

## Abstract

Max Born’s statistical interpretation made probabilities play a major role in quantum theory. Here we show that these quantum probabilities and the classical probabilities have very different origins. Although the latter always result from an assumed probability measure, the first include transition probabilities with a purely algebraic origin. Moreover, the general definition of transition probability introduced here comprises not only the well-known quantum mechanical transition probabilities between pure states or wave functions, but further physically meaningful and experimentally verifiable novel cases. A transition probability that differs from 0 and 1 manifests the typical quantum indeterminacy in a similar way as Heisenberg’s and others’ uncertainty relations and, furthermore, rules out deterministic states in the same way as the Bell-Kochen-Specker theorem. However, the transition probability defined here achieves a lot more beyond that: it demonstrates that the algebraic structure of the Hilbert space quantum logic dictates the precise values of certain probabilities and it provides an unexpected access to these quantum probabilities that does not rely on states or wave functions.

## 1. Introduction

The Boolean algebra (or the equivalent Boolean lattice) is a mathematical structure playing an important role in many scientific and technical fields such as logic, probability theory, circuitry, computer science. Only quantum theory challenges the general applicability of this structure, since the dichotomic observables (those with spectrum $\left\{0,1\right\}$) do not form a Boolean algebra, but a lattice where the distributivity law fails [1,2,3].

The system of the dichotomic observables is called the *quantum logic* and becomes the framework for a new general definition of the *transition probability*. This definition includes not only the well-known quantum mechanical transition probabilities between pure states or wave functions, but further physically meaningful and experimentally verifiable novel cases. It is pointed out that the transition probabilities have a purely algebraic origin, which has mostly been ignored in the past. In this way, they become very different from the classical probabilities which result from probability measures.

Other approaches to the transition probabilities are possible; one is based on projective quantum measurement (Lüders–von Neumann quantum measurement process) [4] and another is based on a non-Boolean extension of the conditional probabilities [5]. The approach presented here, based on the new definition, is more elementary than these two since it does not require advanced concepts such as quantum measurement or the non-Boolean conditional probabilities. Some basic knowledge of quantum mechanics and linear algebra (here particularly the Cauchy–Schwarz inequality) are enough to understand the paper.

After a brief sketch of the quantum logic and its state space in Section 2, the transition probability will be defined formally in Section 3. In Section 4, some examples will be studied to reveal the link to the well-known quantum mechanical transition probabilities and to identify some further novel cases. The connection to the typical quantum indeterminacy is discussed in Section 5.

## 2. Quantum Logic and States

Commonly, the quantum mechanical observables are mathematically represented by self-adjoint (Hermitian) linear operators on a Hilbert space *H*. The dichotomic observables (those with spectrum $\left\{0,1\right\}$) become self-adjoint projection operators and form the *quantum logic*
$L_{H}$. It includes 0 and the identity I. By considering the one-to-one relation between the self-adjoint projection operators and the closed linear subspaces of *H*, it becomes evident that $L_{H}$ is a lattice with order relation ≤, infimum ∧ and supremum ∨. Moreover, 0 is the smallest element, I is the largest element in $L_{H}$ and, for any $p\in L_{H}$, $p^{\prime }:=I-p$ is the orthogonal complement of *p*.

For $p,q\in L_{H}$, p≤q is equivalent to each one of the following conditions: pq=p, qp=p, pqp=p or qpq=p. A pair $p,q\in L_{H}$ is called *orthogonal* if $p\leq q^{\prime }$ or if one of the following equivalent conditions holds: $q\leq p^{\prime }$, pq=0, qp=0, pqp=0 or qpq=0. Orthogonality means that *p* and *q* mutually exclude each other.

A *state* shall allocate a probability to each element of the quantum logic in a consistent way and thus becomes a map $\mu :L_{H}\to \left[0,1\right]:=\left\{s\in R:0\leq s\leq 1\right\}$ with μ(I)=1 and μ(p+q)=μ(p)+μ(q) for each orthogonal pair $p,q\in L_{H}$ [1]. The states form a convex set $S(L_{H})$ which is called the *state space*.

If the dimension of the Hilbert space is two, it is necessary to distinguish between $S(L_{H})$ and the subset $S_{lin}\left(L_{H}\right)$; $S_{lin}\left(L_{H}\right)$ consists of those states that can be extended to a linear map defined for all bounded observables. If it exists, this linear extension is unique because of the spectral theorem and is denoted by μ again. Due to Gleason’s theorem [6,7], the identity $S\left(L_{H}\right)=S_{lin}\left(L_{H}\right)$ holds for all other Hilbert space dimensions (≠2). Although $S_{lin}\left(L_{H}\right)$ is associated with the linear structure of the observables, $S(L_{H})$ depends on the algebraic structure of the quantum logic only.

## 3. Transition Probability

The novel definition of the transition probability shall now be presented. If a pair $p,q\in L_{H}$ with p≠0 and some r∈[0,1] satisfy the identity
$$\mu (q)=r\;\mathrm{for}\;\mathrm{all}\;\mu \in S_{lin}\left(L_{H}\right)\;\mathrm{with}\;\mu (p)=1,$$

*r* is called the *transition probability from*
*p* to *q* and is denoted by P(q|p). The identity P(q|p)=r then becomes equivalent to the set inclusion
$$\left\{\mu \in S_{lin}\left(L_{H}\right):\mu \left(p\right)=1\right\}\subseteq \left\{\mu \in S_{lin}\left(L_{H}\right):\mu \left(q\right)=r\right\}$$
and means that whenever the probability of *p* is 1, the probability of *q* is determined and must be *r*. Particularly in the situation after a quantum measurement that has provided the outcome *p*, the probability of *q* becomes *r*, independently of the initial state before the measurement.

If $0\neq p_{2}\leq p_{1}$ and $P(q|p_{1})$ exists with $p_{1},p_{2},q\in L_{H}$, then $P(q|p_{2})$ exists and $P\left(q|p_{1}\right)=P\left(q|p_{2}\right)$. This follows immediately from the above definition.

The transition probability P(q|p) is a characteristic of the algebraic structure of the observables. If the Hilbert space dimension does not equal two, we have $S\left(L_{H}\right)=S_{lin}\left(L_{H}\right)$ and the transition probability becomes a characteristic of the even more basic structure of the quantum logic.
**Theorem** **1.***Suppose that p≠0 and q are elements in the quantum logic $L_{H}$.**(i)* *The transition probability from p to q exists and P(q|p)=r iff the linear operators p and q satisfy the simple algebraic identity*pqp=rp.*(ii)* *If p≠0≠q holds and if both transition probabilities P(q|p) and P(p|q) exist, they are equal:*P(q|p)=P(p|q).
**Proof.** (i) The linear extension of a state μ becomes a positive linear functional and the Cauchy– Schwarz inequality holds:
$$\left|\mu (xy)\right|\leq {\left(\mu (x^{2}\right)}^{1/2}{\left(\mu (y^{2}\right)}^{1/2}$$
for all bounded linear operators *x* and *y* [8]. This implies that for $p\in L_{H}$ with μ(p)=0, μ(xp)=μ(px)=0 for all bounded linear operators *x*. Please note that projections are idempotent ($p^{2}=p$).⇐: Suppose pqp=rp. If $\mu \in S_{lin}\left(L_{H}\right)$ and μ(p)=1, then $\mu (p^{\prime })=0$ and $\mu \left(q\right)=\mu \left(pqp\right)+\mu \left(p^{\prime }qp\right)+\mu \left(pqp^{\prime }\right)+\mu \left(p^{\prime }qp^{\prime }\right)=\mu \left(pqp\right)=r\mu \left(p\right)=r$. Therefore, P(q|p)=r.⇒: Now suppose P(q|p)=r. Let μ be any state in $S_{lin}\left(L_{H}\right)$. If μ(p)=0, then μ(pqp)=0=μ(rp). If μ(p)>0, define a state $\mu_{p}\in S_{lin}\left(L_{H}\right)$ by $\mu_{p}\left(x\right)=\mu \left(pxp\right)/\mu \left(p\right)$ for $x\in L_{H}$. Then $\mu_{p}\left(p\right)=1$ and therefore $r=\mu_{p}\left(q\right)=\mu \left(pqp\right)/\mu \left(p\right)$. We now have μ(pqp)=rμ(p) for all $\mu \in S_{lin}\left(L_{H}\right)$ and thus pqp=rp.(ii) Suppose that p≠0≠q holds and that P(q|p) and P(p|q) both exist. By (i) we have that $pqp=r_{1}p$ and $qpq=r_{2}q$ with $r_{1}=P\left(q|p\right)$ and $r_{2}=P\left(p|q\right)$. Then $r_{1}pq=pqpq=r_{2}pq$ and either $r_{1}=r_{2}$ or pq=0. In the second case, pqp=0=qpq and therefore $r_{1}=0=r_{2}$ □.

An immediate consequence of the theorem is that P(q|p)=1 iff p≤q and that P(q|p)=0 iff *p* and *q* are orthogonal.

The transition probability is invariant under unitary transformations *u*: P(q|p) exists if and only if $P(uqu^{-1}|upu^{-1})$ exists, and $P\left(q|p\right)=P\left(uqu^{-1}|upu^{-1}\right)$. This follows from the above theorem and directly from the definition of the transition probability.

If the transition probability P(q|p) exists, one can use the *trace* (*tr*) to calculate it: $tr\left(pq\right)=tr\left(p^{2}q\right)=tr\left(pqp\right)=P\left(q|p\right)tr\left(p\right)$ and therefore
P(q|p)=tr(pq)/tr(p).

The term tr(pq)/tr(p) always exists (unless p=0), but it represents a transition probability as defined above only in certain cases. The trace and the last equation cannot help to identify these cases.

There is an interesting connection between the transition probability defined here and the *equiangularity* studied in [9]: if the projections *p* and *q* are equiangular, the transition probabilities P(q|p) and P(p|q) both exist. Transition probabilities are not considered in [9], but this follows by combining theorem 2.3 from there with the above theorem.

## 4. Examples

**Example** **1.**
*Suppose that p≠0 and q commute and that P(q|p)=r exists. Then pqp=rp and $rpq={\left(pq\right)}^{2}=pq$ and either pq=0 or r=1. In the first case, p and q are orthogonal and P(q|p)=0. In the second case, we have p≤q and P(q|p)=1. This means that a non-trivial transition probability (0<P(q|p)<1) requires that p and q do not commute.*


**Example** **2.**
*Suppose that $|\psi \rangle$ is a normalized element of the Hilbert space H. With the common bracket notation (Dirac notation), $p:=|\psi \rangle \langle \psi |$ becomes the projector on the one-dimensional subspace generated by $|\psi \rangle$. For any other $q\in L_{H}$ we then have $pqp=|\psi \rangle \langle \psi |q|\psi \rangle \langle \psi |=\langle \psi |q|\psi \rangle p$. This means that P(q|p) exists for all elements $q\in L_{H}$ with $P\left(q|p\right)=\langle \psi |q|\psi \rangle$. The map q→P(q|p) becomes the pure state defined by $|\psi \rangle$.*


**Example** **3.**
*If q, too, is the projector on a one-dimensional subspace and if this subspace is generated by the normalized element $|\phi \rangle \in L_{H}$, we have $q:=|\phi \rangle \langle \phi |$ and*
$$P\left(q|p\right)={\left|\langle \psi |\phi \rangle \right|}^{2}.$$
*This is the well-known quantum mechanical transition probability between the Hilbert space elements $|\psi \rangle$ and $|\phi \rangle$, often known as wave functions or pure states.*


However, the general and abstract definition of the transition probability in Section 3 goes beyond this situation. The following example demonstrates that the existence of P(q|p) does not require *p* to be a projector on a one-dimensional space.

A non-zero transition probability P(q|p)=r≠0 requires that the dimension of the image qH of *q* is not smaller than the dimension of the image pH of *p*. This can be seen in the following way: the identity pqp=rp with r≠0 implies that pqp and *p* have the same image pH=pqpH⊆pqH; therefore dim(pH)≤dim(pqH)≤dim(qH).

If 0<P(q|p)<1, then $P\left(q^{\prime }|p\right)=1-P\left(q|p\right)\neq 0$ and the dimensions of the images of both *q* and its orthogonal complement $q^{\prime }$ cannot be smaller than the dimension of the image of *p*. Therefore, a case with 0<P(q|p)<1 and dim(pH)>1 requires that the dimension of the Hilbert space *H* is not less than four.

**Example** **4.**
*Consider the matrices*
$$p:=\left(\begin{array}{cccc} 1 & 0 & 0 & 0\\ 0 & 1 & 0 & 0\\ 0 & 0 & 0 & 0\\ 0 & 0 & 0 & 0 \end{array}\right)\;and\;q:=\left(\begin{array}{cccc} {s_{1}}^{2}+{s_{2}}^{2} & 0 & s_{1}s_{3} & -s_{2}s_{3}\\ 0 & {s_{1}}^{2}+{s_{2}}^{2} & s_{2}s_{3} & s_{1}s_{3}\\ s_{1}s_{3} & s_{2}s_{3} & {s_{3}}^{2} & 0\\ -s_{2}s_{3} & s_{1}s_{3} & 0 & {s_{3}}^{2} \end{array}\right)$$
*with $s_{1},s_{2},s_{3}\in R$ and ${s_{1}}^{2}+{s_{2}}^{2}+{s_{3}}^{2}=1$. Some matrix calculations show that $p^{2}=p$, $q^{2}=q$ (i.e., $p,q\in L_{H}$) and that $pqp=\left({s_{1}}^{2}+{s_{2}}^{2}\right)p$. The theorem then yields that P(q|p) exists with*
$$P\left(q|p\right)={s_{1}}^{2}+{s_{2}}^{2}=1-{s_{3}}^{2}.$$


Many further examples can be constructed by using $upu^{-1}$ and $uqu^{-1}$ instead of *p* and *q* with any unitary transformation *u*; then $P\left(uqu^{-1}|upu^{-1}\right)=P\left(q|p\right)$.

Only if *p* is a projector on a one-dimensional space, the transition probability P(q|p) can be represented in the familiar way with the inner product of the Hilbert space as in the examples 2 and 3. In the general case, however, this is not possible; nevertheless, P(q|p) constitutes the physically meaningful and experimentally verifiable probability of *q* in the situation after a quantum measurement that has provided the outcome *q*. The initial state before the measurement becomes irrelevant here in the same way as with a projector *p* on a one-dimensional space.

**Example** **5.**
*With the same p as in the last example, P(q|p) exists only for some, but not for all $q\in L_{H}$. Since we have dim(pH)=2, the above dimension considerations in connection with example 4 show that P(q|p) cannot exist for any q that is a projector on a one-dimensional subspace and not orthogonal to p.*

*If dim(pH)≠1, P(q|p) exists only for some, but not for all $q\in L_{H}$ and, therefore, the definition of a state by q→P(q|p) fails.*


**Example** **6.**
*A case, where P(q|p) exists, but P(p|q) does not exist, can be constructed by using any projector p on a one-dimensional space and any projector q with dim(qH)≥2 and pq≠0 (i.e., p and q are not orthogonal). Since dim(pH)=1, P(q|p) exists, and since dim(pH)<dim(qH), P(p|q) cannot exist.*


## 5. Quantum Indeterminacy

Our common sense, philosophy, logic and the classical sciences make us think that each proposition is either true or false; there is nothing in between. Is it possible to allocate an attribute ‘true’ or ‘false’ in a consistent way to each element of the quantum logic $L_{H}$? Replacing ‘true’ by 1 and ‘false’ by 0, this would result in a state μ with $\mu \left(p\right)\in \left\{0,1\right\}$ for all $p\in L_{H}$. Such a state is called *deterministic* (or *dispersion-free*). However, the Bell-Kochen-Specker theorem tells us that a deterministic state is impossible on the Hilbert space quantum logic $L_{H}$ except that the dimension of the Hilbert space is two [10,11,12].

This can also be seen by considering the transition probabilities. Suppose that 0<P(q|p)<1 holds for $p,q\in L_{H}$ and that μ is a deterministic state. Since μ(p)=1 would imply $\mu \left(q\right)=P\left(q|p\right)\notin \left\{0,1\right\}$, it follows that μ(p)=0. We thus get μ(p)=0 for all projectors *p* on one-dimensional subspaces and, furthermore, for all orthogonal sums of such projectors. With a finite-dimensional Hilbert space *H*, this would exhaust all elements of the quantum logic $L_{H}$ including I and yield a contradiction to μ(I)=1. Therefore, $S_{lin}\left(L_{H}\right)$ does not include any deterministic state and, for dim(H)≠2, there is no deterministic state at all.

However, the transition probability P(q|p) achieves more than just ruling out determinism. It dictates the precise value of the probability of *q*, whenever *p* carries the probability 1 (particularly in the situation after a quantum measurement that has provided the outcome *q*), and is a characteristic of the algebraic structure of the quantum logic.

The most famous manifestation of the typical quantum indeterminacy is Heisenberg’s *uncertainty relation* [13]. Furthermore, general uncertainty relations are due to Robertson [14] and Schrödinger [15]. A transition probability P(q|p)=r with 0<r<1 also represents a kind of uncertainty relation: if *p* is known with certainty, *q* must be unknown and carry the probability *r*.

## 6. Conclusions

It is well-established that the algebraic structure of the Hilbert space quantum logic rules out deterministic states (Bell-Kochen-Specker theorem [10,11,12]). In the present paper, it has been seen that the algebraic structure does a lot more beyond that; it dictates the precise values of the transition probabilities which thus provide an unexpected access to quantum probability that does not rely on states or wave functions.

If $p\in L_{H}$ and $q\in L_{H}$ commute, only three cases are possible for the transition probability: either it does not exist or P(q|p)=1, which is equivalent to p≤q, or P(q|p)=0, which is equivalent to the orthogonality and mutual exclusivity of *p* and *q* (example 1). The same holds, if *p* and *q* were elements of a Boolean algebra, where p≤q defines a logical relation and means that the proposition *p* implies the proposition *q*. Therefore, P(q|p) can be considered an extension of this logical relation to the Hilbert space quantum logic. This extended relation, however, is associated with a probability and introduces a continuum of new cases between the two classical cases ‘*p* implies *q*’ and ‘*p* rules out *q*’.

The transition probabilities introduced here include not only the well-known quantum mechanical transition probabilities between pure states or wave functions, but further physically meaningful and experimentally verifiable novel cases where P(q|p) exists although *p* is a projection on a subspace with dimension higher than one (Example 4). These novel cases become possible also in quantum logics that do not contain any projections on one-dimensional subspaces. Such quantum logics are formed by the self-adjoint projections in the von Neumann algebras of the types II and III, which play an important role in quantum field theory and quantum statistical mechanics [16].

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
