# Peer review of "Quantum Probability’s Algebraic Origin"

_entropy, 2020, doi:10.3390/e22111196_

Round 1

Reviewer 1 Report

The author proposes novel definition of the transition probability. It refers to the algebra of dichotomic  observables without considering some particular states. Some properties of the new notion are derived and a number of examples discussed. In the case of onedimensional  projectors  one obtains the standard formula for quantum mechanical transition probability provided the projectors are identified with relevant pure states. However, for mixed states the situation is more complicated. In general, author’s definition is not applicable while the standard quantum transition probability can be computed.

Concluding, the relevance of author’s  proposal should be more extensively discussed before the paper is considered for publication.

Author Response

Thank you for the review report. I appreciate your comments. Please see the attachment

Reviewer 2 Report

This is an interesting paper which presents some useful arguments about quantum uncertainty. I believe it should be published in its present form.

This paper provided an algebraic approach to the use of probability in setting up the mathematics of quantum theory.   The algebra is well-constructed and rigorous and the paper provides an interesting argument in the foundations of quantum theory.

Author Response

Thank you very much for this review.

Round 2

Reviewer 1 Report

With the additional explanations provided by the author I can recommend the paper for publication.